# Generating Airborne Ultrasonic Amplitude Patterns Using an Open Hardware Phased Array

Rafael Morales [1], Iñigo Ezcurdia [2], Josu Irisarri [2], Marco A. B. Andrade [3] and Asier Marzo [2,*]

1 UltraLeap Ltd., Bristol BS2 0EL, UK; rafael.morales@ultraleap.com
2 UpnaLab, Public University of Navarre, 31006 Pamplona, Spain; inigofermin.ezcurdia@unavarra.es (I.E.); josu.irisarri@unavarra.es (J.I.)
3 Institute of Physics, University of São Paulo, São Paulo 05508-090, Brazil; marcobrizzotti@gmail.com
* Correspondence: asier.marzo@unavarra.es; Tel.: +34-948-16-9715

**Abstract:** Holographic methods from optics can be adapted to acoustics for enabling novel applications in particle manipulation or patterning by generating dynamic custom-tailored acoustic fields. Here, we present three contributions towards making the field of acoustic holography more widespread. Firstly, we introduce an iterative algorithm that accurately calculates the amplitudes and phases of an array of ultrasound emitters in order to create a target amplitude field in mid-air. Secondly, we use the algorithm to analyse the impact of spatial, amplitude and phase emission resolution on the resulting acoustic field, thus providing engineering insights towards array design. For example, we show an onset of diminishing returns for smaller than a quarter-wavelength sized emitters and a phase and amplitude resolution of eight and four divisions per period, respectively. Lastly, we present a hardware platform for the generation of acoustic holograms. The array is integrated in a single board composed of 256 emitters operating at 40 kHz. We hope that the results and procedures described within this paper enable researchers to build their own ultrasonic arrays and explore novel applications of ultrasonic holograms.

**Keywords:** acoustic hologram algorithm; open ultrasonic array; acoustic tweezers

## 1. Introduction

The ability to produce dynamic ultrasonic fields with target shapes is of fundamental importance in ultrasonic imaging [1], nondestructive testing [2,3], and high-intensity focused ultrasound HIFU therapy [4]. When operating in air, there are numerous emerging applications that require the generation of acoustic fields with certain shapes, such as non-contact tactile feedback [5–7], volumetric displays [8,9], parametric audio generation [10,11], and the contactless manipulation of objects [12–16].

In recent years, optical holographic methods have been adapted to acoustics [13,16–18], opening the possibility of generating arbitrary acoustic fields that can be controlled in real time. Acoustic holography is normally achieved using either passive metamaterial structures [17,19,20] or an array of ultrasonic transducers [13,16,21]. Metamaterial structures have the main advantage of allowing for the generation of acoustic fields with a higher spatial resolution, but they cannot dynamically change the field. In contrast, phased arrays do not have this limitation, since the emission phase and amplitude of each transducer can be controlled by a computer, allowing to change the acoustic field in real time. This capability of phased arrays is encapsulated in commercially available platforms, e.g., Ultraleap, Bristol, UK; Pixie Dust Tech., Tokyo, Japan; SonicEnergy, California, USA, each of which provides technology development and commercialisation towards specific target market solutions. Despite the numerous scientific advancements made in both industry and academia, there is currently no unifying hardware platform that can flexibly support exploratory research in acoustic holography applications.

In this paper, we present SonicSurface, a low-cost open hardware array for generating arbitrary acoustic fields in mid-air. We also present an algorithm for calculating the emission amplitude and phase for each transducer in order to create a target amplitude field at a certain distance from the array. Additionally, we offer a comparison of the accuracy of the generated fields depending on the size of the ultrasonic emitters as well as their phase and amplitude resolution. This paper is accompanied by video instructions, available at www.upnalab.com (accessed on 20 February 2021), Do-it-Yourself. Given the low-cost and the use of off-the-shelf components, we hope that researchers can build and use these ultrasonic arrays for their own experiments. We also note the companies commercializing ultrasonic phased arrays offer proprietary solutions that are certified for their use in various commercial applications.

## 2. Related Work

Our review of related work gives an overview of projects that designed and built ultrasonic arrays that typically operate at 40 kHz. Additionally, we provide a review of algorithms for creating an arbitrary pressure field.

An ultrasonic phased array consists of a collection of elements that can transmit or receive ultrasonic waves with specific time delays (phases offsets) and amplitudes. This technology enables the generation of arbitrary pressure fields by controlling the phases and amplitudes of each emitter. Moreover, it provides an interesting setup for a wide spectrum of novel applications, such as mid-air displays [22], wireless power transfer [23], acoustic imaging [24], or delivering food through acoustic levitation [25], to mention a few.

Iwamoto et al. first demonstrated ultrasonic mid-air haptic feedback [26], who developed a prototype consisting of 12 annular channels with a total of 91 ultrasound transducers in a hexagonal arrangement, a single focal point could be refocused along the central axis perpendicular to the array. Shinoda's group [27–30] developed a more sophisticated system that was capable of controlling individually 249 transducers, being able to focus at different 3D positions in space, their boards have the capability to be chained to operate as a larger array system. Carter et al. [6] developed a phased array that can produce multi-point haptic feedback. Ultraleap (Bristol, UK) is a company that commercializes ultrasonic phased arrays for haptic applications related to automotive [31], digital signage [32], and AR/VR [33] applications. The company has also been exploring the effects on humans of high intensity ultrasound exposure [34] and has been releasing multiple prototypes that explore optimized array designs [35,36]. For example, transducer array in a Fibonacci spiral arrangement can suppress unwanted secondary focal points [37]. Pixie Dust Technologies (Tokyo, Japan) provides a parametric speaker [10] and an acoustic levitator [38] based on ultrasound phased arrays. The parametric prototype array has 269 transducers populating a circular array, $\pi/32$ phase resolution, and can be refreshed at 1 kHz. Their levitator prototype has four orthogonally placed phased arrays with 285 transducers with a phase resolution of $\pi/8$ and it is updated at 1 kHz. These ultrasound phased arrays have a fast update rate, high-power output, and sufficient phase and amplitude resolution; however, they are comparatively expensive, the software is closed, and the hardware cannot be easily modified.

Some researchers have developed open platforms of acoustic phased arrays operating at 40 kHz in air. These platforms allow developers to create their own low-cost array [39–41]. For example, TinyLev [42] is a single-axis acoustic levitator that uses two ultrasonic arrays facing each other, reducing the number of independent channels by arranging transducers within the same distance to the trapping positions. Hirayama et al. [9] presented an acoustic levitator display with two opposed arrays that was capable of creating and modulating a large number of focal points at high speeds (20 kHz update rate) for delivering tactile feedback and parametric audio at the same time. While some part of the code is public, the hardware was not provided. Other projects have released both the hardware and software. For example, Ultraino [41] is a multi-purpose phased array that is accompanied by a platform that helps designers to build small phased arrays. The

hardware is based on an Arduino MEGA microcontroller and provides 64 channels with $\pi/5$ phase resolution. Furthermore, multiple boards can be chained together, expanding the number of individual controlled channels. The software is capable of customising phased array arrangements and visualising the pressure field in real-time. Despite the advantage of being a low-cost platform, the operating voltage is limited, reaching a large number of channels is cumbersome, and the transducers need to be wired to the boards. This last part gives some flexibility, but it makes the setups complicated to build, even when just flat geometries are required.

A more detailed review of the available ultrasonic phased arrays can be found in [40]. We reckon that the presented hardware, SonicSurface, provides the most affordable and simple flat phased-array. More importantly, within this paper, we provide an algorithm that is capable of generating arbitrary acoustic fields using SonicSurface or other arrays that provide phase control.

Acoustic holography [43] involves obtaining the near field of a radiating surface by taking measurements on the far field. It is a fundamental technique in health structure monitoring or mechanical vibration analysis. During the last years, a new trend in acoustic holograms has emerged [13,16–18], which is defined as the application of techniques, previously used in optics, to obtain target acoustic fields of different shapes by engineering the amplitude and phase of an array of emitters or an emission modulating surface.

From an algorithmic point of view, researchers first implemented single-point algorithms [5,26] or single traps with different shapes [13]. Later, multi focal-point algorithms [16,44,45] enabled creating high-amplitude points at independent positions. For example, Plasencia et al. [46] proposed a method for optimizing the phases and amplitudes of the acoustic field, obtaining higher-quality points than previous phase-optimization approaches.

Other strategies used a phase modulation plate on top of a flat radiating piston. Melde et al., used an iterative algorithm [17] in order to calculate the required phase modulation to create a target field at a given distance; they employed a static 3D printed modulator that encoded the phases for reconstructing the target hologram. Brown et al. [47] introduced a second holographic plate to modulate both phase and amplitude surface.

These algorithms assume a high-resolution modulation plate with almost pixel-like shape for each point that modulates the field. Differently, here we introduce a modification on the previous algorithms to obtain target amplitude fields using discrete ultrasonic arrays that are made of circular emitters.

## 3. Hardware Design

SonicSurface is a phased array consisting of 256 transducers emitting at 40 kHz. The transducers are arranged in a $16 \times 16$ grid and built on a single integrated printed circuit board (PCB). On one side of the PCB, ultrasonic emitters are soldered, whereas, on the other side, the field-programmable gate array (FPGA) (EP4CE6E22C8N—ALTERA IV Core Board, Waveshare), shift registers (74HC595, TI), drivers (MIC4127 from MT), and decoupling capacitors (ceramic 50V 0.1 µF) are mounted. The signals for each emitter are generated by the FPGA. The shift registers demultiplex each digital line coming from the FPGA into eight channels, and the drivers boost the voltage of the channels from logic voltage to the supplied power voltage (up to 20 peak-to-peak voltage (Vp-p)). A block diagram can be seen in Figure 1.

The calculation of the phases and amplitudes to be emitted is performed on an external computer and then sent to the FPGA via Serial Universal asynchronous receiver-transmitter (UART) protocol at 203,400 bauds. A double buffer has been implemented in the FPGA to generate the signals uninterruptedly [48]; one of the buffers stores emission patterns coming from the computer, whereas the second buffer is the one that is used by the FPGA to continuously generate the emission signals, a command from the computer swaps the buffers at once. This method avoids latency and waiting issues. Different versions of the firmware are available for the FPGA to support phase and amplitude control, or amplitude modulation of the 40 kHz main signal.

The protocol used to communicate with the FPGA is presented in Table 1; 1 byte specifies commands or emission patterns. If the byte value is larger than 127, it is a command; otherwise, it represents an emission phase offset or amplitude, depending on the mode. By default, the FPGA has a resolution of 32 divisions per period, so numbers from 0 to 31 represent phases from 0 to $2\pi$, 32 represents no emission. Receiving a value of 254 indicates that new phases are going to be sent, the read pointer of the buffer is set to channel 0; afterwards, each phase sent will be assigned into the current read pointer and the pointer increased by one. The command 253 indicates swapping of the buffers. Other commands are: 252, to toggle amplitude modulation at 200 Hz for haptic feedback applications; 252 indicates that instead of phases, amplitudes are going to be sent. From 192 to 196, indicates the board number to activate (being 192 board number 1), in the case that multiple boards were chained together.

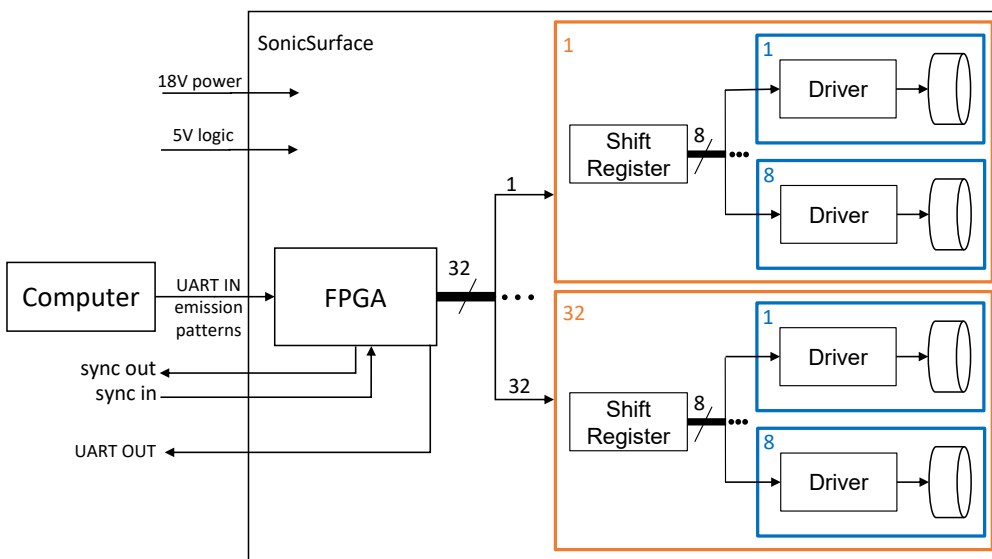

**Figure 1.** Schematic of the SonicSurface ultrasonic array. A field-programmable gate array (FPGA) receives the phases to be emitted from a computer, they are stored on a double buffer and constantly output. The FPGA multiplexes 8 channels into one line so that only 32 output pins are needed. There are 32 blocks of shift registers, being able to drive a total of 256 emitters.

The FPGA can generate 256 square-wave signals at 40 kHz. Each of the signals supports a phase delay control of 32 divisions per period or $\pi/16$ radians, the amplitude can be modulated with up to 16 divisions. A multiplexing scheme strategy was employed for reducing the number of needed output pins and, thus, reduce the price of the FPGA. Packs of eight channels are multiplexed into one digital line. Later, this line is demultiplexed back into eight channels while using the shift registers. Figure 2 illustrates the channel multiplexation scheme from the FPGA and circuit implementation.

**Table 1.** Communication protocol commands.

| BYTE | Command | Action |
| --- | --- | --- |
| 0XXX XXXX | Set phases or amplitudes | Sets the phase or amplitude for the current channel and moves the pointer to the next channel |
| 1111 1110 | Start receiving phases | Sets the current channel to the first one and set that values will represent phases |
| 1111 1101 | Swap buffer | Swaps the emission and read buffer |
| 1111 1100 | Toggle modulation | A modulation of 200 Hz on the amplitude |
| 1111 1100 | Amplitude mode | Sets the current channel to the first one and set that values will represent amplitudes |

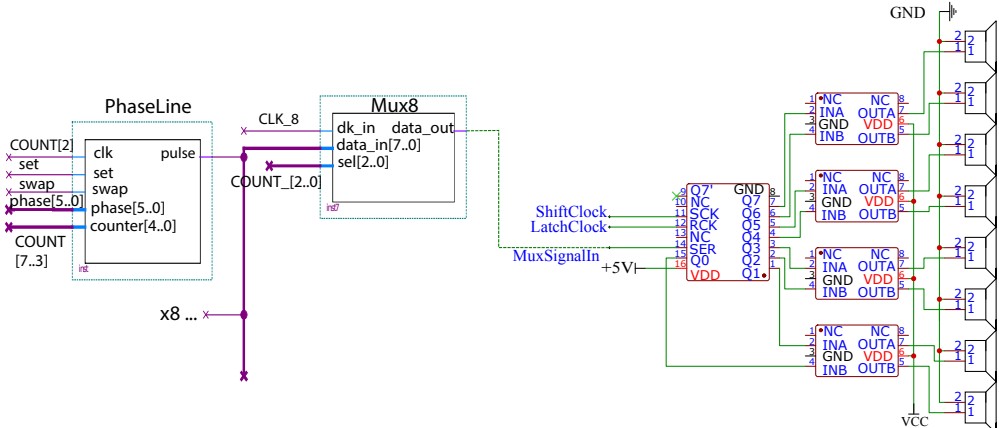

**Figure 2.** At the left, the FPGA blocks in charge of generating the signals are presented, 8 phaseLine blocks (signal generators) are multiplexed into one digital line to reduce the required number of output pins. At the right, the circuit schematic represents a shift register that demultiplexes the signal into 8 channels that get amplified by four dual Metal–oxide–semiconductor field-effect transistor (MOSFET) drivers and fed into the ultrasonic emitters.

The shift and the latch clock are generated by the FPGA. The shift clock controls when the shift registers shift data in, the latch clock determines when the data that were shifted should be output. The shift clock operates at 10.24 MHz (8 multiplexed channels × 40 kHz × 32 divisions per period), whereas the latch clock operates at 1.28 MHz (40 kHz × 32 divisions per period). The number of divisions per period (i.e., the resolution in phase or amplitude) could be doubled to 64, but the shift clock would operate slightly above 20 MHz, which would require better filtering and traces on the PCB.

Once the digital signal for each channel has been demultiplexed, it is amplified from 5 V up to 20 V using a dual Metal–oxide–semiconductor field-effect transistor (MOSFET) driver (e.g., TC4427a or MIC4127 from MT). After testing different electronic components for amplifying the signals (e.g., L293D or BJT transistors), MOSFET drivers were found to efficiently drive the ultrasonic transducers. Dual Mosfet Drivers can amplify two channels and have a small footprint; larger components would not fit on the integrated board. Subsequently, the output of the drivers is fed into the ultrasonic emitters (a comparison of suitable transducers can be found in the supplementary information of TinyLev [42]). Given the narrowband nature of the emitters, it is possible to use a half-square wave to drive them without generating a significant amount of harmonics [41]. This technique is widely employed for airborne ultrasonic phased arrays, because generating a digital square signal is less complex than creating an analog sinusoidal signal, they are also easier to amplify.

We present two models of the ultrasonic array. In the first one, the electronic components (i.e., shift registers, drivers, and decoupling capacitors) are surface mounted device (SMD) and the ultrasonic emitters have a diameter of 10 mm (Figure 3). The second model uses emitters of 16 mm diameter and through-hole (TH) components (Figure 4). The first model is more compact and faster to assemble if SMD equipment is available (e.g., stencils, solder paste, and a reflow oven). The TH model is larger and it takes more time to assemble, but it can be done with entry level electronics equipment (i.e., a soldering iron). Throughout the paper, we focus our experiments on the SMD board, since we think that it will be employed more often in the scientific community.

The program synthesized for the FPGA delegates the phase calculations on an external computer, thereby the cost of the board itself can be kept low. A UART reader block gets the bytes coming from the external computer [49]. A distributor block stores the current channel and sets the phases on the 256 signal generator blocks, each generator block outputs a digital signal of 40 kHz. Each generator block stores two phases, the one to be emitted and the previously read phase. The generator blocks have an internal amplitude

counter that represents the number of divisions that the output should be HIGH, there is a global counter (from 0 to 31) that reaches all of the blocks, when the phase of a generator block coincides with the global counter, the internal amplitude counter is set to the target amplitude. The generator blocks have a dataline of five bits to read phases or amplitudes when the line setPhase or setAmp goes high. It also has a swap line, which swaps the phases/amplitudes when it goes high; this is to implement the double buffer. Eight generator blocks are grouped into a multiplexer, giving a total of 32 multiplexed lines that are output from the FPGA, as well as the shift and latch clocks. There are six auxiliary general-purpose inputs/outputs (GPIOs) (we have denominated them from A to F) that can be operated, defined, and implemented by the user. For example, B is used as the UART input, D is used as sync out (internal 40 kHz reference), E can be used as sync in (40 kHz signal to synchronize the global counter), and A can be used as a UART out; C and F are free for custom applications. Figure 5 shows the block diagram of the FPGA firmware.

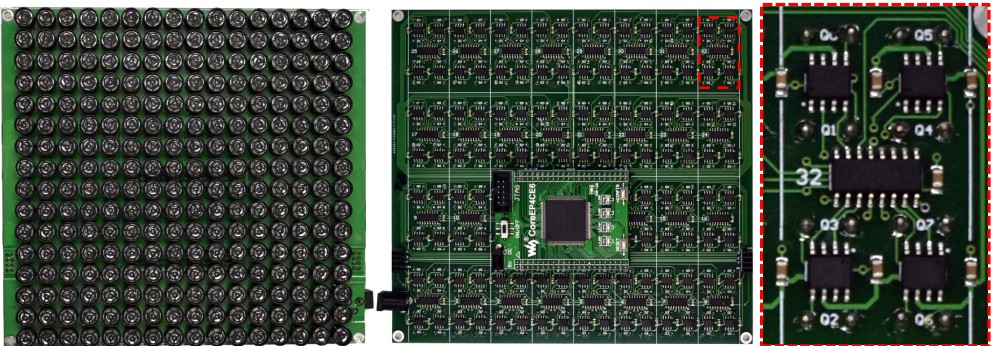

**Figure 3.** Board with surface mounted devices and emitters of 10 mm diameter. (**left**) Top view of the Sonic surface where $16 \times 16$ ultrasonic emitters can be seen. At the sides there are connectors for power, Universal asynchronous receiver-transmitter (UART) in, grounds, sync out and sync in. (**center**) bottom view where the shift register blocks can be seen with the FPGA on top. (**right**) closer view on a shift register block where a shift register demultiplexes a digital line into eight signals that are fed to four dual-drivers and then into eight ultrasonic emitters.

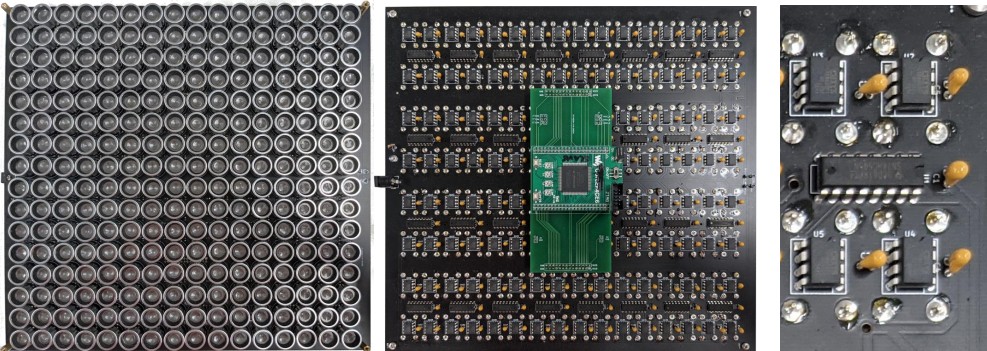

**Figure 4.** Ultrasonic array built with Through-hole components and emitters of 16 mm diameter. (**Left**) transducers of 16 mm diameter soldered on the printed circuit board (PCB). (**Center**) back of the board with the shift registers, drivers and decoupling capacitors. The FPGA board is connected through an expander board. (**Right**) detailed view of a shift register block.

The UART Reader and Distributor blocks operate with the internal clock, the generator blocks and multiplexers operate with a clock that is synchronized with the sync in signal. Thereby, when multiple boards operate together, the emission waves have exactly the same frequency. If the emission clocks were not synchronized, traveling waves would be created [41], making the generation of static fields impossible. A master board has its sync out connected to its sync in, slave boards take the sync signal from the master board.

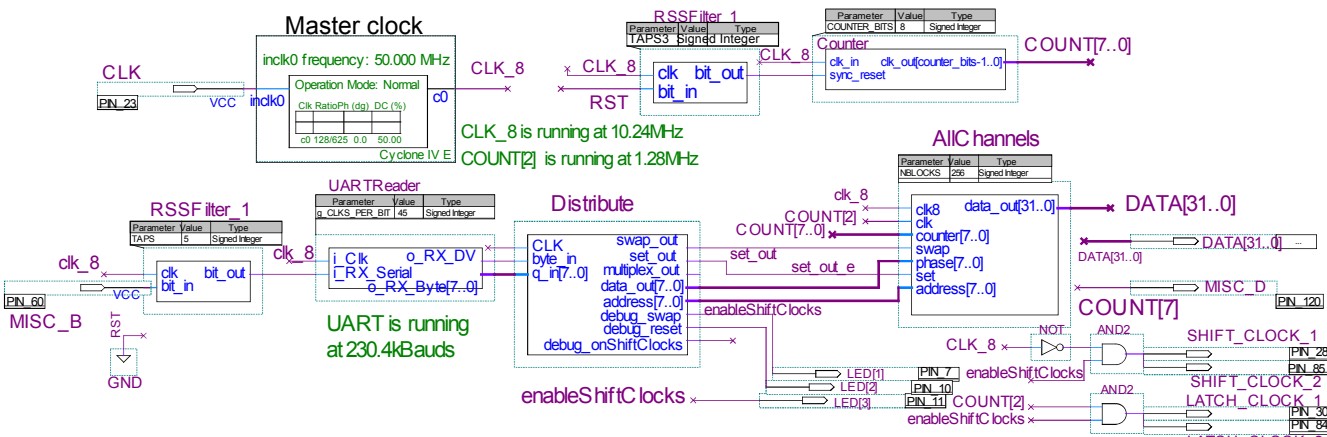

**Figure 5.** Block diagram of the code that is synthesized in the FPGA. On the top-left, the MasterClock is a phase-locked loop (PLL) to transform the internal 50 MHz clock into a 10.24 MHz clock named CLK_8. At the top-right, there is a global counter that acts as a frequency divider of CLK_8: COUNT[7] sets at 40 kHz and is output as the reference signal on MISC_D; COUNT[2] is the latch clock. If the board acts as a slave, the counter is synchronized with a 40 kHz external signal filtered by a RSS filter. On the bottom left, the UART input is filtered, read, and sent to the distributor. The distributor updates the emission phases of the generator blocks. AllChannels contain 256 generator blocks that connect to 32 Multiplexers of eight channels each. The generator blocks and multiplexers are timed by the outputs of the global counter. At the bottom-right, the multiplexed data channels as well as the latch and shift clocks are output.

The presented hardware has been optimized for an operating frequency of 40 kHz. This is the most common frequency for airborne ultrasonic phased arrays [9,38,41,42], operating at higher frequencies is not straightforward. On the one hand, the multiplexation of signals is used to reduce the required traces on the PCB and pins on the FPGA, our current system requires just a two-layer PCB and 40 GPIOs of the FPGA. However, this multiplexation leads to a 10.24 MHz shift clock. Increasing the frequency or phase resolution would require a higher clock frequency, which is beyond what is recommended for a simple PCB or the specs of the shift registers. On the other hand, commercially available transducers that operate at higher frequencies (e.g., 100 kHz or 400 kHz from MultiComp) are 10 mm in diameter and, thus, emit a very narrow beam. The emission from an array of these emitters would not interfere between each other and, thus, would not be suitable for the techniques presented here or phased-array techniques in general.

## 4. Algorithm

The algorithm that was developed by Melde et al. [17] is a modification of the Gerchberg–Saxton algorithm [50]. It calculates the phase modulation necessary at each point in the emitter plane in order to obtain a target amplitude field at the desired distance. The issue is that the algorithm is designed to produce modulation profiles that are almost continuous with more than $100 \times 100$ elements that are smaller than half-wavelength. However, available airborne ultrasonic arrays have a resolution of $16 \times 16$ or $24 \times 24$ at most, with element sizes that are larger than the wavelength and a circular shape instead of a square. We introduced a modification on this algorithm to consider the discrete nature of ultrasonic arrays and their lower number of elements when compared to passive modulators.

The proposed algorithm is an iterative approach with four steps per iteration, as described in Figure 6. The FOCUS library was employed for the forward propagation and the backpropagation of the emission and target field slices [51].

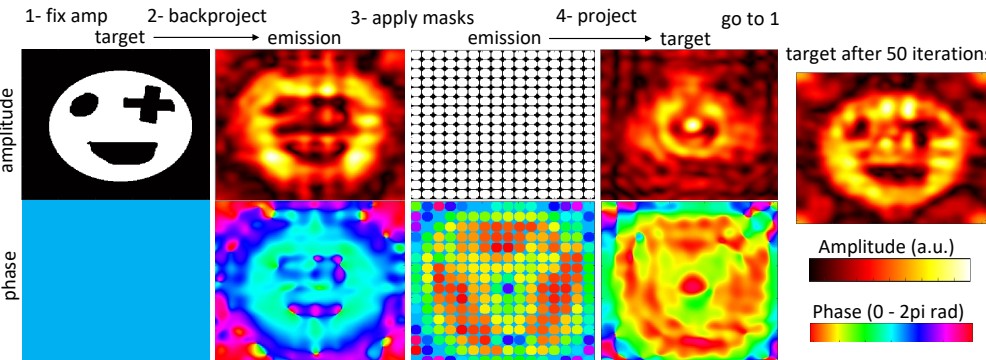

**Figure 6.** Iterative algorithm to determine the emission phases and amplitudes for an array of emitters. Step (1) fix the amplitude into the target slice, the phase is not modified. Step (2) Backproject the target into the emission. Step (3) Apply on the emission slice a discretization on phase, amplitude, and spatial resolution, as well as the mask with the shape of the emitters. Step (4) Project the emission into the target. After 50 iterations of steps 1 to 4, the target amplitude is shown at the left.

## 5. Results

### 5.1. Comparison between Simulations and Experiments

The experimental setup of Figure 7 was used to measure the acoustic pressure distribution generated by the array in order to compare the emitted experimental amplitude slices with the simulated ones. In this setup, an ultrasonic receiver (MA40S4S, Murata) is attached to the head of a delta stage (Anycubic Kossel) and the emitter array sits on its bed. A Matlab script communicates with the delta stage and it moves the receiver to different positions on a grid of 16 × 16 cm with 2.5 mm spacing. At each measuring point, the computer reads the peak-to-peak voltage that was captured by the oscilloscope (Hantek 6074BE). The voltage is linearly proportional to the amplitude and, thus, can be directly translated to amplitude in arbitrary units (a.u.). The computer sends the emission phases to the array through the UART protocol and it controls the stage using the G-Code protocol. Figure 8 shows the obtained experimental amplitude slices, which are in reasonable qualitative agreement with the simulation slices, except for the Brazilian flag pattern.

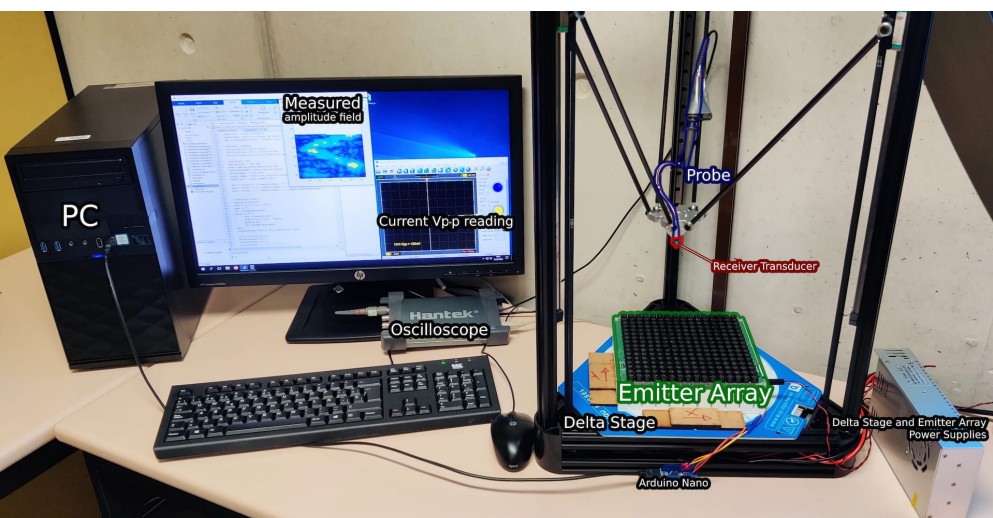

**Figure 7.** Experimental Setup used to scan the emitted amplitude slice.

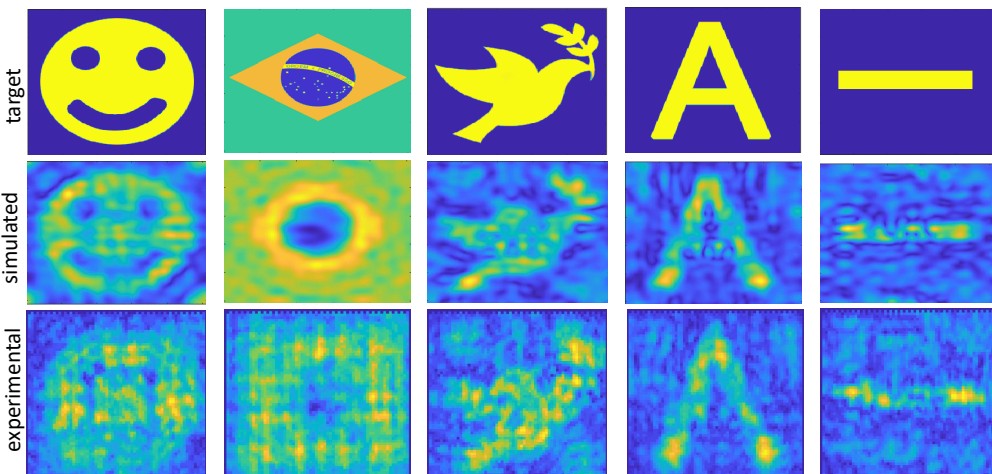

**Figure 8.** Amplitude slices obtained for different patterns, plotted using the function imagesc of Matlab. The first row is the target, the second one is the simulated slice, and the third row is the experimental measurement.

### 5.2. Effect of Phase, Amplitude, and Spatial Resolution

We carried out multiple simulations using the algorithm that is described in Section 4 with different parameters for emitter size, phase emission resolution and amplitude emission resolution. All of the target amplitude fields were generated 16 cm above the array, since we tested that, at that distance, the best results were obtained. The default simulation parameters are those from the SMD board, i.e., *emitterSize* = 10 mm, *phaseResolution* = 32, and no amplitude modulation. One parameter was varied at a time and the mean square error (MSE) of the obtained imaged was obtained. The results can be seen in Figure 9.

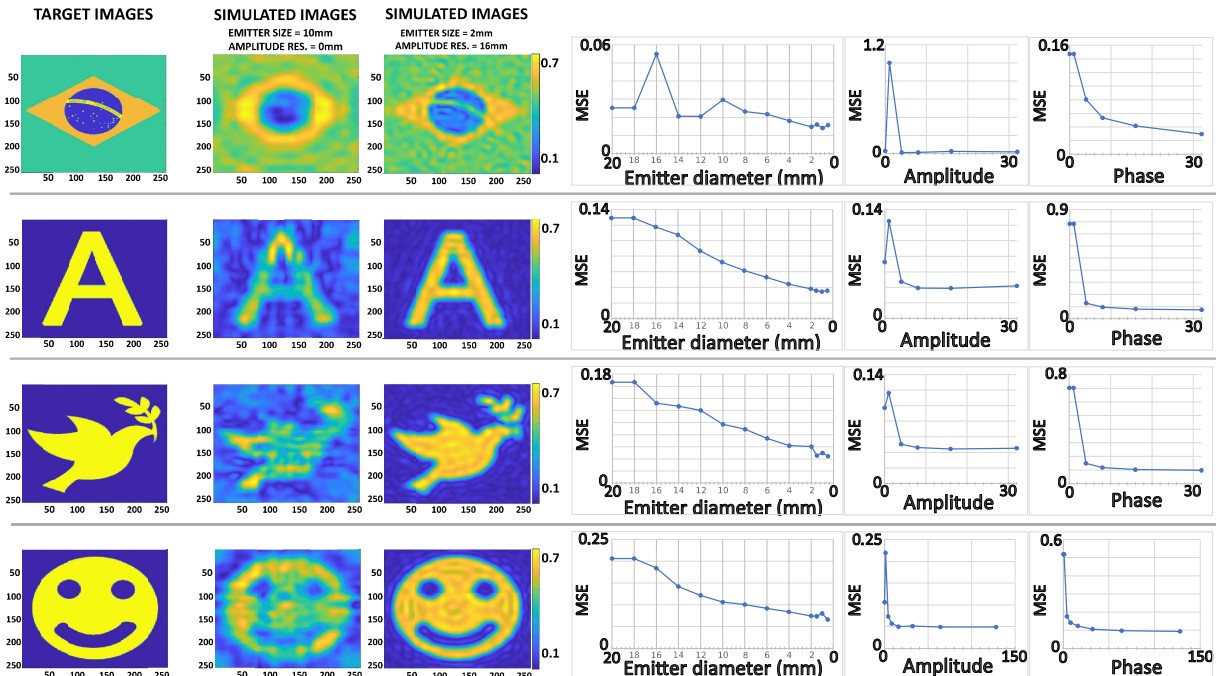

**Figure 9.** Simulated amplitude fields at 16 cm from the array for different target patterns and array parameters. (First column) target amplitude field. (Second column) obtained amplitude field when the emitter array is the surface mounted device (SMD) board presented in the paper, i.e., *emitterSize* = 10 mm (transducer diameter), *phaseResolution* = 32 and no amplitude modulation. (Third column) obtained amplitude fields with an array with *emitterSize* = 2 mm, *phaseResolution* = 32 and *amplitudeResolution* = 16. (Fourth column) mean square error (MSE) as the emitter size decreases. (Fifth column) MSE as phase resolution increases, *emitterSize* = 10 mm. (Sixth column) MSE as the amplitude resolution increases, *emitterSize* = 10 mm.

The patterns employed were: the flag of Brazil (non-binary image), the letter A, a Dove, and a smiley face. In general, it can be seen that as the emitter size decreases (i.e., more spatial resolution), the quality of the images improves. It is important to note that significant reductions of MSE are obtained, even when emitters get smaller than half-wavelength (4.3 mm), and that no further improvement is obtained below 2 mm (1/4 of the wavelength); this is different from the generation of regular focal points that do not increase its amplitude once the emitters are reduced below half-wavelength size [13]. The phase resolution significantly improves the pattern quality, but quickly plateaus when the phase resolution reaches eight divisions per period; this is in accordance with the simulations performed for simple focal points [41]. For amplitude resolution, it is clear that having amplitude modulation reduces the MSE by half even when only four different amplitudes can be emitted. In summary, the sweet-spot is obtained with a phase resolution of eight divisions per period and amplitude resolution of four divisions; the MSE improves as the emitters get smaller (i.e., more spatial resolution), but no improvement is found once the emitter size reaches quarter-wavelength.

These findings could be specific to the patterns that were selected in the study and to our setup characteristics (e.g., wavelength, number of emitters or distance to the target slice); however, the code was made public, so that other researchers could run simulations for their specific setups (e.g., operating in water or with static metamaterials).

## 6. Conclusions

In recent years, Acoustic holography has found numerous applications and has advanced rapidly due to the adaptation of methods found in the optics community. In this paper, we have attempted to advance, test, and unify algorithms and hardware used for acoustic mid-air holography. Namely, we have described a novel iterative algorithm that calculates the emission phases and amplitudes for an array of emitters that can be used to generate a desired target amplitude field. To our knowledge, this is the first algorithm capable of determining the amplitude and emission phases for discrete arrays comprised of finite sized emitters. We have then used this algorithm to investigate the effects of increased phase, amplitude and spatial resolution in the obtained amplitude field. Our analysis demonstrates that diminishing returns are observed at some point on-wards. Meaning that depending on the application requirements there is no need to use expensive hardware or that the computations can be accelerated by further discretizing the solution domain. Finally, to support the growth of the acoustic holography research community, we have described an open hardware platform named SonicSurface which is an affordable FPGA-based ultrasound phased array. Two different models for the array of emitters have been provided (SMD and TH), so that researchers from different fields and backgrounds can customise these further for their own experimental requirements. We hope that the algorithm and hardware presented in this paper facilitates further research on the field of ultrasonic arrays and enables novel applications of crafted amplitude fields.

**Author Contributions:** Conceptualization, A.M. and R.M.; software, A.M., I.E. and J.I.; investigation, R.M., A.M., I.E. and J.I.; writing—original draft preparation, M.A.B.A., R.M., A.M. and I.E.; writing—review and editing, M.A.B.A., R.M., A.M., I.E. and J.I.; supervision, M.A.B.A. and A.M. All authors have read and agreed to the published version of the manuscript.

**Funding:** This research was funded by the Government of Navarre (FEDER) 0011-1365-2019-000086 and from the European Union's Horizon 2020 research and innovation programme under grant agreement No 101017746, TOUCHLESS.

**Data Availability Statement:** No new data were created or analyzed in this study. Data sharing is not applicable to this article.

**Acknowledgments:** We thank Adrian Vicente for the calibration of the scanning stage. We thank Joshua Taylor, Euan Freeman and Chi Thanh Vi for building this array for their research.

**Conflicts of Interest:** The authors declare no conflict of interest.

## Abbreviations

The following abbreviations are used in this manuscript:

| | |
|---|---|
| FPGA | A field-programmable gate array |
| PLL | Phase-locked loop |
| HCI | Human-Computer Interaction |
| HIFU | High-intensity focused ultrasound |
| 3D | Three-dimensional |
| Hz | Hertz is the derived unit of frequency in the International System of Units (SI) |
| kHz | Kilohertz |
| MHz | Megahertz |
| PCB | Printed circuit board |
| UART | Universal asynchronous receiver-transmitter |
| PLL | Phased locked loop |
| PWM | Pulse width modulation |
| I/O | Input and Output |
| GPIOs | A general-purpose inputs/outputs |
| V | Voltage |
| MOSFET | Metal–oxide–semiconductor field-effect transistor |
| SMD | Surface mounted device |
| TH | Through-hole |
| MSE | Mean square error |
| MT | Mounted |
| Vp-p | Peak-to-peak voltage |
| a.u. | Arbitrary unity |
| mm | Millimetre |
| cm | Centimetre |
| rad | Radian |
| uF | Microfarad |

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
