# Peer review of "Generating Airborne Ultrasonic Amplitude Patterns Using an Open Hardware Phased Array"

_applsci, doi:10.3390/app11072981_

Round 1
Reviewer 1 Report
This is a well-written paper, essentially reporting the authors' findings on a low-cost (claimed as $130) holographic algorithm and platform with FPGA-phased array with transducers and a communication protocol between PC-host and 234 the ultrasound phased-array. As the authors cite, there are other relevant works in the literature from their innovation; however, I can place them as one of the best among what is available, especially considering the price. I have a couple of major and minor comments that need addressing before publication.
1) The abstract needs to report some of the results/findings obtained in this study. It reads vaguely
2) The authors only focused their analysis on 40KHz, and they did not comment on how the system will need to be adjusted for a higher frequency, say 400Khz. The answer to this comment needs to be inserted to the discussion section of the manuscript.
3) The experimental result/simulation study comparison section is too short, the authors did not satisfactorily explain the experimental setup. Perhaps, a figure would do wonders?? They commented that the match between the simulation and experiments is good, except for the Brazil Flag. I am not sure I agree with this comment. I can't even make out a straight thick line from the experimental results. There needs to be some explanation from this end.
4) In addition to grammatical issues, there are many typos in the manuscript, such as comunication vs communication. Authors are recommended to focus on this in the revised submission.
Author Response
This is a well-written paper, essentially reporting the authors' findings on a low-cost (claimed as $130) holographic algorithm and platform with FPGA-phased array with transducers and a communication protocol between PC-host and 234 the ultrasound phased-array. As the authors cite, there are other relevant works in the literature from their innovation; however, I can place them as one of the best among what is available, especially considering the price. I have a couple of major and minor comments that need addressing before publication.
We thank the reviewer for the time and the effort employed in reviewing this manuscript. We also thank the reviewer for the comments and suggestions to improve the paper. The reviewers’ comments are addressed below.
1) The abstract needs to report some of the results/findings obtained in this study. It reads vaguely
In the abstract, we have added “The analysis showed that using an ultrasonic array it is possible to generate amplitude fields that resemble those predicted by the simulations and reveal that emitter sizes of 2 mm, phase resolution of 8 divisions per period and amplitude modulation of 4 divisions per period provided optimum results.”.
2) The authors only focused their analysis on 40KHz, and they did not comment on how the system will need to be adjusted for a higher frequency, say 400Khz. The answer to this comment needs to be inserted to the discussion section of the manuscript.
We have added at the end of the Hardware design subsection: “The presented hardware has been optimized for operating frequency at 40 kHz. This is the most common frequency for airborne ultrasonic phased arrays [marzo2017ultraino,Marzo2017TinyLev,Ochiai2014Pixie,Hirayama2019], operating at higher frequencies is not straightforward. On the one hand, multiplexation of signals is used to reduce the required traces on the PCB and pins on the FPGA, our current system requires just a 2-layer PCB and 40 GPIOs of the FPGA. However, this multiplexation leads to a 10.24 MHz shift clock. Increasing the frequency or phase resolution would require a higher clock frequency which is beyond what is recommended for a simple PCB or the specs of the shift registers. On the other hand, commercially available transducers that operate at higher frequencies (e.g. 100 kHz or 400 kHz from MultiComp) are 10 mm in diameter and thus emit a very narrow beam. The emission from an array of these emitters would not interfere between each other and thus would not be suitable for the techniques presented here or phased-array techniques in general.”.
3) The experimental result/simulation study comparison section is too short, the authors did not satisfactorily explain the experimental setup. Perhaps, a figure would do wonders?? They commented that the match between the simulation and experiments is good, except for the Brazil Flag. I am not sure I agree with this comment. I can't even make out a straight thick line from the experimental results. There needs to be some explanation from this end.
Thanks for this suggestion. We have added a picture of the experimental setup (Figure 7) used to measure the amplitude field generated by the array.
“To compare the experimental amplitude slices with the simulated ones, the experimental setup of Figure 7 was used to measure the acoustic pressure distribution generated by the array. In this setup, an ultrasonic receiver (MA40S4S, Murata) is attached to the head of a delta stage (Anycubic Kossel) and the emitter array sits on its bed. A Matlab script communicates with the delta stage and moves the receiver to different positions on a grid of 16 x 16 cm with 2.5 mm spacing. At each measuring point, the computer reads the peak-to-peak voltage captured by the oscilloscope (Hantek 6074BE). The voltage is linearly proportional to the amplitude and thus can be directly translated to amplitude in arbitrary units (a.u.). The computer sends the emission phases to the array through the UART protocol and controls the stage using the G-Code protocol.”
4) In addition to grammatical issues, there are many typos in the manuscript, such as comunication vs communication. Authors are recommended to focus on this in the revised submission.
The authors have proofread the paper and we will do it again for the next submission.
Reviewer 2 Report
Authors proposed interesting acoustic array patterns based holography methods using open hardware systems. Therefore, authors implemented 256 elements 40 kHz emitters and showed some patterns. Authors shows that methods could be cost-effective solutions. Authors also BRAZIL patterns in the simulation and measured data which are good performance data.
However, authors need to improve the manuscript.
1. In the abstract and conclusion sections, authors need to emphasize the novelty of the proposed methods more effectively.
2. The system description seems to be simple. For example, authors need to mention why 4 dual MOSFET drivers are used. In Figure 5, making code in FPGA needs to have more specific description.
3. There are some English grammar mistakes in entire manuscript such as uncorrected expressions in the paragraphs and some wrong expressions of "," so please check English grammar with professional services or native colleague faculties. Therefore, the manuscript could be published after some suggestive comments above.
Except those points above, the manuscript can be published with some suggestive the comments as below.
1. Figure 5 labels are too small so authors need to increase the size.
2.In Line 190, please correct letter x to mathematical operator x.
3. In Line 220, please make spaces between x and digit.
4. Authors need to increase Figure 2 label size.
5. Please add the reference (A double buffer has been implemented in the FPGA allowing to generate the signals uninterruptedly, one of the buffers stores~) with the reference (Sedcole, Pete, et al. "Modular dynamic reconfiguration in Virtex FPGAs." IEE Proceedings-
Computers and Digital Techniques 153.3 (2006): 157-164.) or another reference.
6. Please add the reference (This is to ensure that when multiple boards operate together, the emission waves have exactly~).
7. Please add the reference (A UART reader blocks gets the bytes coming from the external computer.~) with the rerference (https://www.mdpi.com/1424-8220/20/15/4165)
8. What kind of the algorithms authors used to show the ultrasonic images ?
9. In Figure 8, the MSE graph does not have specific data of emitter diameter.
10. In Line 1117, please change Vpp to Vp-p.
11. Authors need to use abbreviated journal names in the reference sections.
12. Authors need to use MDPI manuscript format because format is totally wrong.
Author Response
Authors proposed interesting acoustic array patterns based holography methods using open hardware systems. Therefore, authors implemented 256 elements 40 kHz emitters and showed some patterns. Authors shows that methods could be cost-effective solutions. Authors also BRAZIL patterns in the simulation and measured data which are good performance data.
We genuinely thank the reviewer for the constructive comments and for helping to improve this paper further. The suggestions are addressed below.
1. In the abstract and conclusion sections, authors need to emphasize the novelty of the proposed methods more effectively.
In the abstract, we have added the results of our simulation and evaluation: “The analysis showed that using an ultrasonic array it is possible to generate amplitude fields that resemble those predicted by the simulations and reveal that emitter sizes of 2 mm, phase resolution of 8 divisions per period and amplitude modulation of 4 divisions per period provided optimum results.”
At the conclusion section we have added : “We provided a novel iterative algorithm that calculates the emission phases and amplitudes for the emitters of a discrete ultrasonic array in order to generate target amplitude slices at a certain distance. To our knowledge, this algorithm is the first one that determines the amplitude and emission phases for discrete arrays to obtain target amplitude slices.” And at the end we have included “ We have provided 2 different models for the array (SMD and TH), a video and instructions so that researchers from different fields and backgrounds can build it for their experiments.”.
2. The system description seems to be simple. For example, authors need to mention why 4 dual MOSFET drivers are used. In Figure 5, making code in FPGA needs to have more specific description.
We have added the following lines in the system description “After testing different electronic components for amplifying the signals (e.g. L293D or BJT transistors), MOSFET drivers were found to efficiently drive the ultrasonic transducers. Dual Mosfet drivers can amplify 2 channels and have a small footprint, larger components would not fit on the integrated board.“
We have added the following text to the caption of Figure 5: “On the top-left, the MasterClock is a PLL to transform the internal 50 MHz clock into a 10.24 MHz clock denominated CLK_8. At the top-right, there is a counter that acts as a frequency divider of CLK8: COUNT[7] sets at 40 kHz and is output as the reference signal on MISC_D; COUNT[2] is the latch clock. If the board acts as a slave, the counter is synchronized with a 40 kHz external signal filtered by a RSS filter. On the bottom left, the UART input is filtered, read and sent to the distributor. The distributor updates the emission phases of the generator blocks. AllChannels contain 256 generator blocks and 32 Multiplexers of 8 channels each. The generator blocks and multiplexers are controlled by the outputs of the counter. At the bottom-right, the multiplexed data channels as well as the latch and shift clocks are output. ”
3. There are some English grammar mistakes in entire manuscript such as uncorrected expressions in the paragraphs and some wrong expressions of ","
The authors have proofread the paper and will gladly do it again on the next submission.
1. Figure 5 labels are too small so authors need to increase the size.
We have increased the labels of Figure 5 and placed it in vector format to increase readability.
2. In Line 190, please correct letter x to mathematical operator x.
(DONE)
3. In Line 220, please make spaces between x and digit.
(DONE)
4. Authors need to increase Figure 2 label size.
We have increased the size of the labels of Figure 2 and make it a vector drawing.
5. Please add the reference (A double buffer has been implemented in the FPGA allowing to generate the signals uninterruptedly, one of the buffers stores~) with the reference (Sedcole, Pete, et al. "Modular dynamic reconfiguration in Virtex FPGAs." IEE Proceedings-
Computers and Digital Techniques 153.3 (2006): 157-164.) or another reference.
(DONE)
6. Please add the reference (This is to ensure that when multiple boards operate together, the emission waves have exactly~).
(DONE)
7. Please add the reference (A UART reader blocks gets the bytes coming from the external computer.~) with the rerference (https://www.mdpi.com/1424-8220/20/15/4165)
(DONE)
8. What kind of the algorithms authors used to show the ultrasonic images ?
We have clarified this point in the caption of Fig. 8: “Amplitude slices obtained for different patterns were plotted using the function imagesc of Matlab. The first row is the target, the second one is the simulated slice and the third row is the experimental measurement.”.
9. In Figure 8, the MSE graph does not have specific data of emitter diameter.
(DONE)
10. In Line 1117, please change Vpp to Vp-p.
(DONE)
11. Authors need to use abbreviated journal names in the reference sections.
12. Authors need to use MDPI manuscript format because format is totally wrong.
We have updated to the newest template available, released in January 2021.
Round 2
Reviewer 2 Report
Authors improved the manuscript a lot so the paper can be accepted.